# Integration of pan-cancer multi-omics data for novel mixed subgroup identification using machine learning methods

**Seema Khadirnaikar[1], Sudhanshu Shukla [ID][2]\*, S. R. M. Prasanna[1]**

**1** Department of Electrical Engineering, Indian Institute of Technology Dharwad, Dharwad, Karnataka, India,
**2** Department of Biosciences and Bioengineering, Indian Institute of Technology Dharwad, Dharwad, Karnataka, India

\* sudhanshu@iitdh.ac.in

**Data Availability Statement:** All the data used in this study can be downloaded from the GDC data portal: https://portal.gdc.cancer.gov/ or from PanCanAtlas Publication: https://gdc.cancer.gov/

## Abstract

Cancer is a heterogeneous disease, and patients with tumors from different organs can share similar epigenetic and genetic alterations. Therefore, it is crucial to identify the novel subgroups of patients with similar molecular characteristics. It is possible to propose a better treatment strategy when the heterogeneity of the patient is accounted for during subgroup identification, irrespective of the tissue of origin. This work proposes a machine learning (ML) based pipeline for subgroup identification in pan-cancer. Here, mRNA, miRNA, DNA methylation, and protein expression features from pan-cancer samples were concatenated and non-linearly projected to a lower dimension using an ML algorithm. This data was then clustered to identify multi-omics-based novel subgroups. The clinical characterization of these ML subgroups indicated significant differences in overall survival (OS) and disease-free survival (DFS) (p-value<0.0001). The subgroups formed by the patients from different tumors shared similar molecular alterations in terms of immune microenvironment, mutation profile, and enriched pathways. Further, decision-level and feature-level fused classification models were built to identify the novel subgroups for unseen samples. Additionally, the classification models were used to obtain the class labels for the validation samples, and the molecular characteristics were verified. To summarize, this work identified novel ML subgroups using multi-omics data and showed that the patients with different tumor types could be similar molecularly. We also proposed and validated the classification models for subgroup identification. The proposed classification models can be used to identify the novel multi-omics subgroups, and the molecular characteristics of each subgroup can be used to design appropriate treatment regimen.

## Introduction

The recent advances in therapy protocols have increased the five-year prognosis of many cancer types, yet, cancer remains the second most common cause of death globally [1]. Classification of cancer is crucial for the primary diagnosis of the disease. Generally, cancers are grouped together based on their organ-of-origin. Clinically, the majority of the tumors with

about-data/publications/pancanatlas. All the codes used in the manuscript are available at https://github.com/seemark11/Pancancer-subgroup-identification.git.

**Funding:** No.

**Competing interests:** The authors have declared that no competing interests exist.

the same histological grade and organ-of-origin are treated using a similar approach [2]. Despite the tumors originating from the same organ and having the matching histopathological grade, patients respond differently to the therapy resulting in different survival outcomes [3]. One of the main reasons for this differential response is the variation in the underlying genetic and epigenetic aberrations that contribute to the heterogeneity [4, 5]. A better treatment regimen can be proposed if the treatment strategies account for this heterogeneity as well [6]. Hence, there is a need to identify the novel subgroups based on the genomic and epigenomic aberrations beyond the organ-of-origin.

The advancements in next generation sequencing technologies have contributed to the generation of a massive amount of data representing various genomic and epigenomic changes. This laid the foundation for the establishment of The Cancer Genome Atlas (TCGA), a multi-platform cancer database housing information of more than 11,000 samples from 33 cancer types. This database instigates us to understand the association of various molecular features with the phenotype. As each molecular level of evidence (omic level or datatype) provides a different set of biomarkers and insights about the aberrations, integrative analysis of multiple datatypes accounting for the non-linear interactions will result in the identification of better subgroups [7, 8]. The idea of accounting for individuals' heterogeneity for the identification of novel subgroups falls in the realm of precision therapy, which intends to use the precise knowledge of the variation in a smaller population's genome to recommend the therapies targeting the mechanisms specific to that sub-population [9]. Several works have attempted to identify the molecular subtypes within the samples from the same tumor type and have shown the presence of multiple subtypes [10–12]. Identification of alterations specific to the subgroups in various studies has resulted in the development of targeted therapies [13, 14].

Samples from different organs-of-origin and different grades can also be similar molecularly. The first attempt to identify the subgroups in pan-cancer data was made by Hoadley et al., using the data from 12 tumor types to identify 11 subgroups [15]. Through their multi-platform analysis, authors showed that the samples from different organs-of-origin did cluster together. This study was based on Cluster-Of-Cluster-Assignments (COCA). COCA is a late integration technique where there is no provision to account for the interaction among the different data types. Recently, another study was carried out by the same group using data from 33 different tumor types [16]. Here, the authors used iCluster to model the interactions within the data types and identified 28 subgroups. However, various studies have pointed out the computational complexity associated with this statistical technique to be quite high [17]. González-Reymúndez et al. proposed a statistical technique based on penalized matrix factorization to identify eight clusters using pan-cancer data (n = 5,408) [18]. Here, the authors have used sparse singular value decomposition (sSVD) to reduce the dimension of data and density-based spatial clustering of applications with noise (DBSCAN) algorithm to cluster the samples. As the dimension of data increases, the computation of SVD gets computationally intensive.

All these pan-cancer studies are based on statistical methods. Statistical models can be used to understand the relation between the variables and draw inferences. However, they cannot be used to identify and generalize the hidden patterns, particularly in the multi-modal heterogeneous data [19]. As the advancement in cost-effective sequencing technologies has enabled the generation of large-dimension data at all levels of the genome, a comprehensive analysis of this data using various machine learning (ML) techniques will help in the identification of hidden patterns. Hence, this work aims to apply ML-based techniques to multi-omics (multiple data types) pan-cancer data for better understanding and subgrouping of cancer patients. Identifying molecularly similar subgroups in pan-cancer samples will help design better therapeutic strategies independent of the organ-of-origin.

In this work, different data types, including mRNA, miRNA, methylation, and protein expression, were integrated. To model the non-linear interaction among the molecular features from multiple data types, an autoencoder (AE) was trained, and a reduced dimension representation was obtained. Consensus $K$-means clustering was then carried out using this representation to identify the ML-based molecular subgroups. Pairwise statistical tests were carried out in the identified subgroups using all the features. Further, the features specific to each subgroup were identified by filtering them based on the q-value and fold-change (FC). The chosen features were then used to train the classification models to identify the multi-omics-based molecular subgroup for a new sample. Three widely used ML-based classification models, support vector machines (SVM), random forest (RF), and feed-forward neural network (FFNN) were trained, and their prediction probabilities were combined to obtain the decision-level fused classification models as the decision-level fused models are more robust and stable to outliers. Also, as each datatype is known to convey complementary information, we trained the classification models by combining the features from different data levels to obtain feature-level fused classification models.

## Materials and methods

The steps followed in this work for the identification of novel ML-based subgroups are outlined in Fig 1 and explained in detail in the subsequent sections.

### Datasets and data preprocessing

The pan-cancer study is a collection of tumor and normal data from 33 different tumor types for multiple data types. RNA-sequencing FPKM values, miRNA RPKM values, DNA methylation 450k beta values, and protein expression values (RPPA data) of solid tumors from TCGA study were used [20–22]. The abbreviation of each cancer type, along with the number of samples having the information for every datatype, is summarized in S1 Table. The samples (n = 5703), which had information from all the datatypes, were considered for further analysis. In this work, the different datatypes are labeled as factors, and the mapping is tabulated in S2 Table. For a machine learning (ML) algorithm to learn effectively and to obtain improved generalizability, the number of samples ($n$) should be approximately equal to the number of dimensions ($p$) (i.e., $n \sim p$) [23, 24]. To satisfy this condition, preprocessing was carried out as outlined in the S1 Fig, following the protocols from the previous studies [12, 25–28]. Briefly, dimensions in $F_1$ and $F_2$ with zeros in more than 20% of the samples were dropped [10]. Dimensions in $F_1$ were then sorted based on the standard deviation in decreasing order, and the top 2000 most variable dimensions were considered for further analysis. In the case of $F_3$, dimensions (probes) with data missing in more than 10% of the samples and those on X and Y chromosomes were dropped. Similar to $F_1$, the dimensions in $F_3$ were sorted based on the standard deviation, and the 2000 most varying probes were considered for further analysis [29]. For $F_4$, proteins with data missing in more than 10% of the samples were dropped. In the case of both $F_3$ and $F_4$, missing values (NAs) were imputed by K-nearest neighbors (KNN) ($K$ = 5) as in previous studies [11, 12, 25]. The selected dimensions were then stacked to obtain the multi-omics data for samples common across all the datatypes. To better understand each subtype, mutation data (maf files) and copy number data (segment data) from the TCGA study were used.

### Multi-omics data integration and cluster identification

The multi-omics data obtained by concatenating different datatypes was further reduced to a lower dimension using autoencoder (AE) to capture the non-linear interaction among the

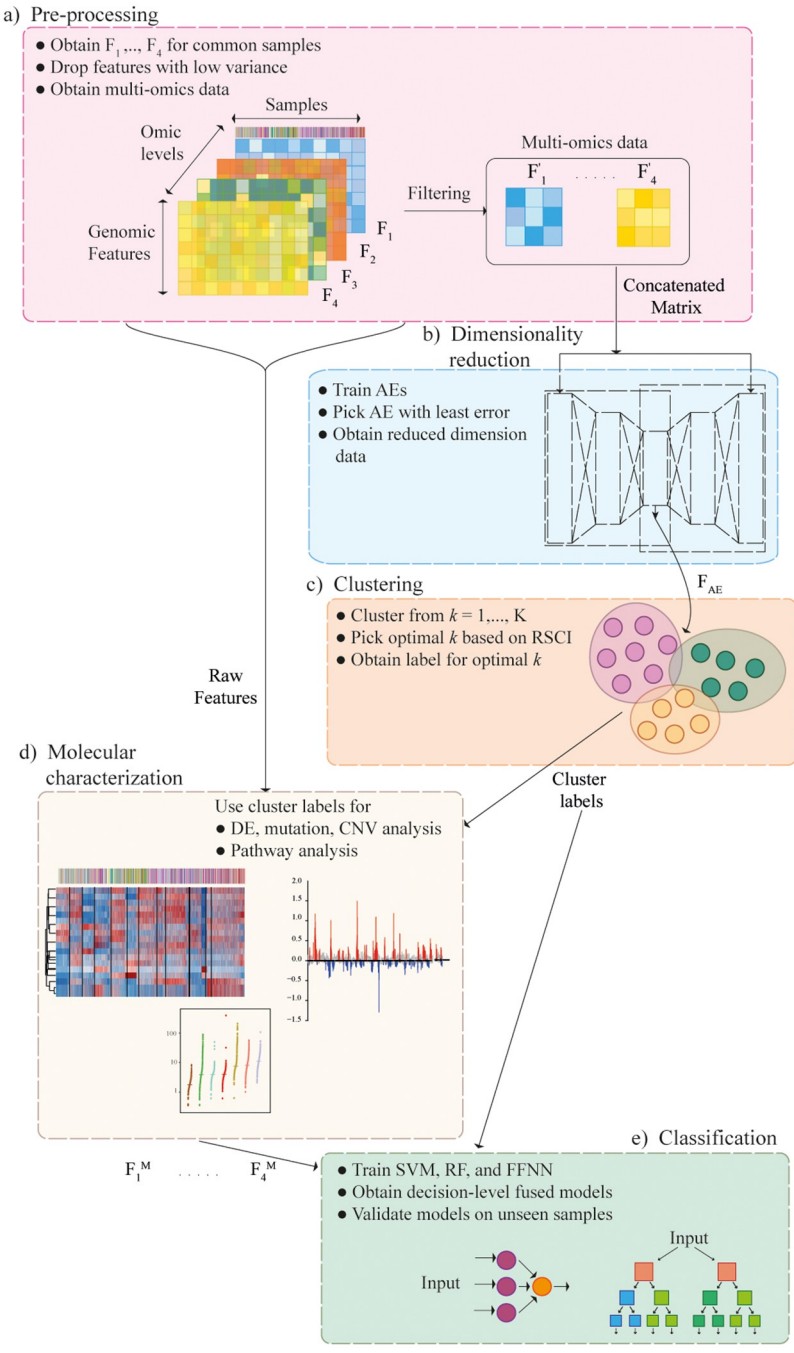

**Fig 1. Overall pipeline followed in this work. (a)** Each data type (single-omic) was preprocessed, and multi-omics representation was obtained by stacking the features for the samples common across all the omic levels. **(b)** The latent representation of multi-omics data ($F_{AE}$) was obtained using an autoencoder (AE). **(c)** Consensus $K$-means clustering was applied on the reduced dimension representation to obtain the cluster labels. **(d)** To understand the subgroups obtained, molecular characterization of samples in each cluster was carried out. **(e)** Decision-level fused classifiers were obtained by the combination of classification models including, support vector machines (SVM), random forest (RF), and feed-forward neural network (FFNN) for multi-omics subgroup identification.

different dimensions [30–32]. The number of nodes in the hidden layers and the bottleneck layer were varied to obtain multiple architectures of AE, which were trained with different learning rates (S3 Table). The AE models were trained with Adam optimizer, and mean-squared error as a loss with early stopping criteria, i.e., model training was stopped if there was no reduction in validation error for ten subsequent epochs. The reduced dimension data from AE was then used to identify the novel subgroups by clustering.

Consensus $K$-means clustering was applied to the reduced dimension data. The number of clusters ($K$) was varied from 2 to 20. Clustering was repeated 1000 times with 80% of the samples chosen randomly [33]. The proportion of samples that cluster together during different runs indicates the consistency of clustering. Proportion of ambiguously clustered pairs (PAC), the value quantified with the aid of the cumulative distribution function (CDF) curve is one such variable [34]. The section lying in between the two extremes of the CDF ($C$) curve ($u_1$ and $u_2$, S2(a) Fig) quantifies the proportion of samples that were assigned to different clusters in each iteration. PAC ($P$) is used to estimate the value of this section representing the ambiguous assignments and is defined by Eq (1), where $K$ is the desired number of clusters.

$$P(K) = C(K, u_2) - C(K, u_1). \tag{1}$$

The lower the value of PAC, the lower the disagreement in clustering during different iterations, or in other words, the more stable the clusters obtained [34]. As PAC does not account for a reference distribution, it is biased towards the higher values of $K$. Relative cluster stability index (RCSI) overcomes this by generating a reference model following the Gaussian distribution without actual clusters (i.e., $K = 1$). The reference data is generated using Monte-Carlo simulation, maintaining the correlation between the dimensions [35]. Hence, RCSI was used in this work to find the optimal number of clusters. For a given number of clusters $K$, RCSI is given by Eq (2).

$$RCSI(K) = \log_{10}\left(\frac{1}{B}\sum_{b=1}^{B} P_{rf}(K, b)\right) - \log_{10}\left(P_{rl}(K)\right), \tag{2}$$

where $B$ is the total number of Monte Carlo simulations, $P_{rf}(K, b)$ is the PAC value of the reference distribution for $b$th Monte Carlo simulation, and $P_{rl}(K)$ is the PAC value for the actual data. Higher the value of RCSI, the better the clustering. In this work, the values of $u_1$ and $u_2$ were set to 0.1 and 0.9, respectively, as suggested by the authors [35].

## Molecular characterization of subgroups

To check the phenotypic differences in the ML subgroups identified, the log-rank test was carried out using overall survival (OS) and disease-free survival (DFS) as endpoints, and the survival difference was visualized using the Kaplan-Meier (KM) curves. Further, to understand and identify the features that define and are specific to each ML subgroup, ANOVA and pairwise t-tests were used. The steps followed while preprocessing and testing each datatype are outlined in S4 Fig. The features with $log_2(FoldChange) \geq 5$ and $q \leq 0.01$ were considered to be differentially expressed and used for further interpretation. To identify the gene ontology (GO) pathways enriched in each ML cluster, Metascape analysis was carried out using the protein-coding genes (PcGs). Also, the expression of the PcGs was used to carry out the Gene Set Enrichment Analysis (GSEA) using the hallmark geneset to better understand the various pathways enriched in each subgroup [36, 37]. The leukocyte fraction and stromal fraction for each sample were also used to gain further insights about the subgroups [16, 38]. To

understand the tumor microenvironment (TME), CIBERSORT analysis was carried out using the LM22 signature gene set [39]. To obtain the mutation signatures, the steps described by the previous studies were followed [16, 40]. To summarize, the mutation signature for each sample was obtained by deconvolution of the mutation matrix with the signature weight matrix by non-negative matrix factorization (NMF) [40]. To understand the copy number variation, the G-score, which accounts for the amplitude of copy number variation and the frequency of occurrence of aberration in each cytoband, was obtained from GISTIC analysis [41]. The G-score was visualized using Maftools [42]. To deduce the pathway activity associated with each ML subgroup, details of 22 gene programs and 20 drug programs were analyzed by plotting their average values [15, 16].

## Subgroup identification by classifier combination

The features specific to each subgroup identified by the molecular analysis in the previous section, along with the subgroup labels obtained by clustering, were used to train the ML models to identify the subgroups for a new sample. Three ML models, support vector machine (SVM), random forest (RF), and feed-forward neural network (FFNN) were trained. The hyper-parameters were tuned by five-fold cross-validation repeated ten times. SVM was trained with a radial kernel as the interaction among the different dimensions is known to be non-linear. FFNN was trained with different learning rates (0.001, 1e-04, and 1e-05), and the optimal one was chosen by a five-fold CV. To obtain the train and test data, the data was split in the ratio of 90%–10%. To obtain the validation set, the training set was further split in the ratio of 90%–10%. Besides the individual classifiers ($L_0$), decision-level fused classifiers ($L_1$) were also built to obtain precise predictions. To train these models, the prediction probabilities obtained from individual classifiers ($P_{SVM}$, $P_{RF}$, and $P_{FFNN}$) were stacked and used as input, and the cluster labels as output. A linear decision-level fused model was obtained by linearly weighing the prediction probabilities from individual $L_0$ classifiers by $\alpha$, $\beta$, and $\gamma$, respectively [30, 43, 44]. The final prediction ($P_F$) was obtained by the weighted summation of individual prediction probabilities using Eq (3) [45].

$$P_F = \alpha \times P_{SVM} + \beta \times P_{RF} + \gamma \times P_{FFNN}. \tag{3}$$

The values of $\alpha$, $\beta$, and $\gamma$ were varied from 0 to 1 in steps of 0.01 by ensuring that they sum up to 1 (S1 File).

Rather than manually iterating through a set of weights to identify the significance of each classifier, ML models like logistic regression and FFNN were also trained using the prediction probabilities from individual classifiers to identify the multi-omics subgroups. As non-linear models were used to combine the prediction probabilities, we called the resulting models as non-linear decision-level fused models.

If the same set of samples are used to train both $L_0$ and $L_1$ classifiers, the performance of the classifier will degrade in the cases where the distribution of the test set is different from the training set. To overcome this, two sets of decision-level fused models, one without a hold-out dataset and one with a hold-out dataset were trained in this work. In the decision-level fused model trained without a hold-out dataset, both $L_0$ and $L_1$ classifiers were trained using all the training samples. For the decision-level fused model with a hold-out dataset, the training dataset was split, and 60% of the data was used to train the $L_0$ models and the rest 40% to train the $L_1$ models.

## Results

### Dimensionality reduction and clustering of pan-cancer data

The multi-omics representation of pan-cancer samples was obtained by concatenation of data from different datatypes (S1 Fig). In a complex disease like cancer, there often exists a non-linear interaction among the different molecular features [46, 47]. To capture this non-linear interaction, multiple autoencoder (AE) models were trained by varying the number of hidden layers and the number of nodes in the hidden layer. The AE model with the least difference in the training and validation loss was chosen to reduce overfitting (S3 Table) [48]. The reduced dimension data from the AE model was then clustered by consensus $K$-means clustering to identify the subgroups across the TCGA tumors. RCSI value, which takes a null reference distribution to calculate PAC was used to pick the optimum number of clusters as described in the methods section. The cluster with the highest RCSI value ($K$ = 13) was considered for further analysis (Fig 2(a) and 2(b), S2(a) Fig). To visualize the distribution of samples, tSNE plots were plotted using the reduced dimension AE data. The samples in the tSNE plots were colored based on the tumor type and the cluster numbers (C1 to C13) (Fig 2(c) and 2(d)). The tSNE plot labeled with 13 ML clusters as identified in this analysis was more compact and well separated (silhouette width = 0.43) than the clusters obtained based on the tumor type (silhouette width = 0.32), showing the superiority of the stratification when carried out using the proposed method.

To validate that dimensionality reduction is essential and the interaction between the molecular features is indeed non-linear, the dimension of the selected raw features was reduced by the most widely used linear dimensionality reduction technique, PCA. The features representing 99% of the variance were retained and used for clustering. The RCSI value obtained for dimensionality reduction by PCA (RCSI = 0.14) was smaller than that obtained by AE (RCSI = 0.8). This indicated that the clusters resulting from the dimensionality reduction by AE were better and more consistent. To confirm that dimensionality reduction is crucial and the clusters obtained using multiple data types are indeed more consistent than clusters obtained using a single data type, we repeated the whole pipeline for each data type using the selected features with and without dimensionality reduction. The resulting RCSI values indicated that dimensionality reduction is an essential step, and the clusters accounting for multiple data types were more stable than single data type (Table 1). Further, we compared our model with iClusterPlus and showed that the clusters obtained using AE based model were more consistent (S4 Table).

### Composition of pan-cancer clusters

To further understand the composition of the ML clusters, we calculated the proportion of samples from different tumor types in each cluster (S3 Fig). Among the 13 ML clusters, six clusters were pure with all the samples from the same tumor type, and the rest were mixed clusters with samples from different tumor types. The ML cluster C1 was purely composed of LGG samples. Similarly, the ML clusters C3, C5, C6, C8, and C12 comprised of samples from THCA, BRCA, PRAD, BLCA, and LIHC, respectively. With samples from 27 different tumor types, ML cluster C13 was the most heterogenous subgroup. This cluster also formed the largest subgroup. ML cluster C10 was the smallest heterogenous cluster, with the majority of the samples from KIRP, CHOL, and LIHC. ML cluster C2 had samples from uterine, breast, and cervical cancers. Hence, we called it GYN cluster. We named ML cluster C4 as kidney cluster as it was formed by the samples from various kidney tumors. COAD, STAD, and READ accounted for the majority of the samples in ML cluster C7 and hence, named as

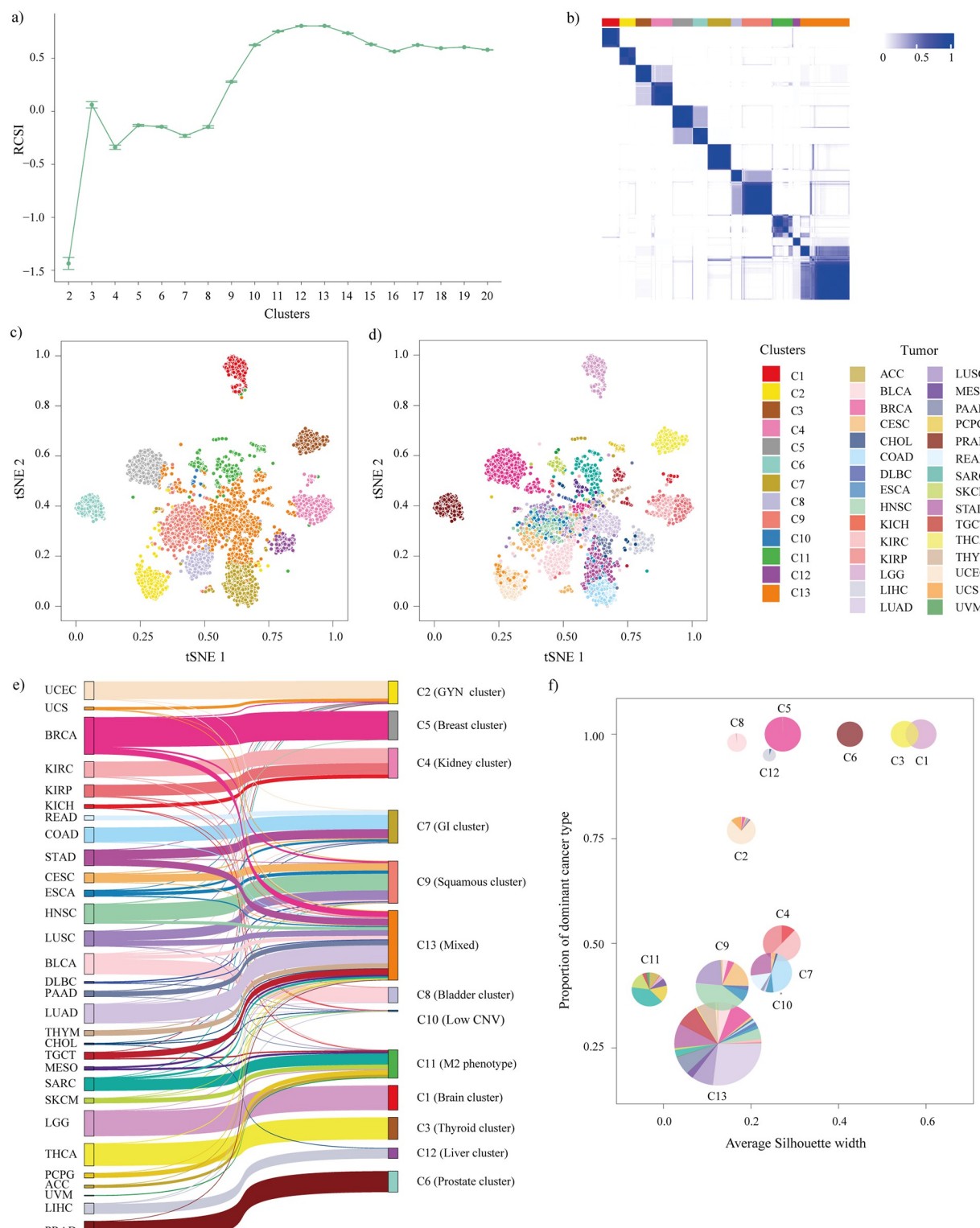

**Fig 2. Composition and distribution of samples in ML clusters. (a)** RCSI values for $K = 2$ to $K = 20$. **(b)** Consensus heatmap plot for consensus $K$-means clustering ($K = 13$). **(c) and (d)** t-SNE plots for reduced dimension obtained using AE. Samples are colored based on the labels obtained by consensus $K$-means clustering and tumor type, respectively. **(e)** Sankey plot depicting the relationship of samples in each cluster with TCGA tumor type. **(f)** Pie chart showing the proportion of samples from various tumor types in each cluster plotted against the average silhouette width showing the compactness of each cluster.

**Table 1. Summarizing the RCSI values obtained for $K = 13$ for each level of evidence for the subset of selected features, when clustered without dimensionality reduction, and with dimensionality reduction using PCA and AE ($F_1$: mRNA expression, $F_2$: miRNA expression, $F_3$: DNA methylation, $F_4$: Protein expression).**

| Omic | Dimension | Without dimensionality reduction | With dimensionality reduction | |
|---|---|---|---|---|
| | | | PCA | AE |
| $F_1$ | 2000 | -0.69 | -0.86 | -0.19 |
| $F_2$ | 1753 | 0.19 | 0.23 | 0.25 |
| $F_3$ | 2000 | 0.35 | -0.45 | 0.43 |
| $F_4$ | 216 | 0.48 | -0.34 | 0.31 |
| $F_1 + F_2 + F_3 + F_4$ | 5969 | 0.27 | 0.14 | 0.8 |

gastrointestinal cluster (GI cluster). Interestingly, samples from HNSC, LUSC, and CESC constituted the majority of samples in ML cluster C9 which was called squamous cluster. ML cluster C11 was a mixed cluster formed by samples from 12 tumor types, with the majority of them from SARC, SKCM, and PCPG. This distribution can be seen in the Sankey plot as well, which depicts the relation between the samples from different clusters and their tumor type (Fig 2(e)).

To understand the compactness and homogeneity within each ML cluster, the proportion of samples from the dominant tumor type in each cluster was plotted against the average silhouette width of each cluster (Fig 2(f)). Here, the radius of each pie is proportional to the number of samples in each cluster. Though the ML clusters dominated by a single tumor type C1 (Brain cluster), C3 (Thyroid cluster) and C6 (Prostate cluster) had the highest silhouette width, ML clusters C4 (Kidney cluster) and C7 (GI cluster), which are mixed clusters, had silhouette widths closer to that of ML cluster C5 (Breast cluster), which is formed by a single tumor type. This result strengthens the hypothesis that, despite originating from different tumor types, the samples in these ML clusters are close to each other molecularly.

## Clinical and biological characterization of clusters

To analyze if there exists any differences in the survival times (overall survival (OS) and disease-free survival (DFS)) between the 13 ML clusters obtained, the log-rank test was carried out to compare the survival times, and Kaplan-Meier (KM) plots to visualize the survival curves. This analysis indicated that there exists at least one subgroup with significantly different survival time, when compared to the others (OS and DFS p-value<0.0001, S2(b) and S2(c) Fig). With an aim to gain further insights into the distinguishing features of each ML cluster contributing to the variation in the phenotype, statistical tests were carried out as described in the methods section. This analysis identified 2868 protein-coding genes (PcGs), 442 long non-coding RNAs (lnc RNAs), 104 miRNAs, 4872 methylation probes, and 216 proteins to be differentially expressed (DE) (Fig 3 and S5 Table).

To interpret the function associated with various genes identified, PcGs identified within each ML cluster were used to carry out the metascape analysis. The top 5 GO biological processes and the genes associated with them were tabulated (S6 and S7 Tables). ML clusters C5 (Breast cluster), C4 (Kidney cluster), and C7 (GI cluster) had activation of pathways associated with embryonic morphogenesis, circulatory system process, and epithelial cell differentiation, respectively. The genes associated with these GO processes were also used to identify the druggable genes and the drug-gene interaction (S8 Table) using The Drug-Gene Interaction Database (DGIdb) [49]. DGIdb was also used to identify the drugs that can be used to target the DE genes in each ML cluster (S9 Table). Further, GSEA analysis was carried out using the

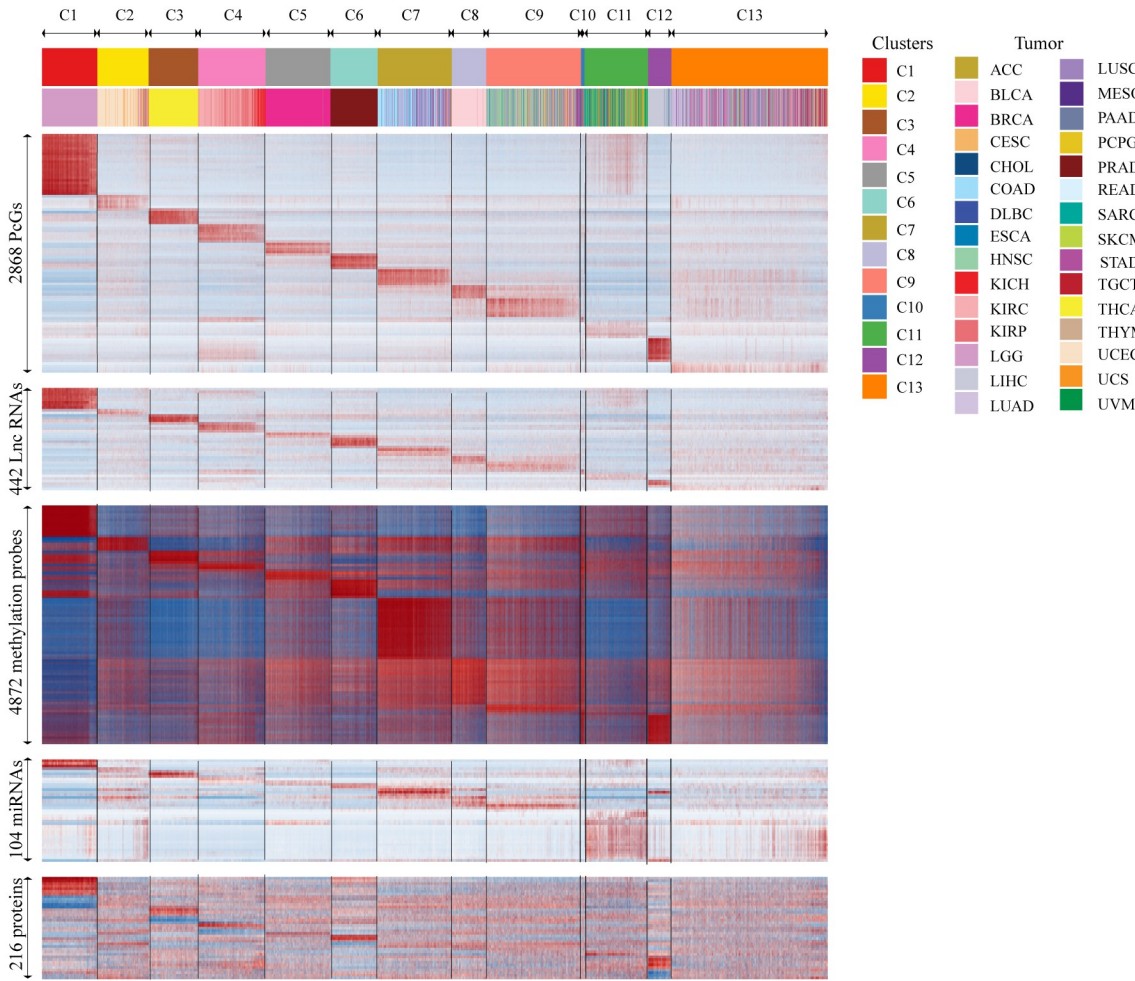

**Fig 3. Characterization of different molecular levels of evidence in ML clusters.** Heatmap indicating the expression of protein-coding genes (PcGs), long non-coding RNAs (Lnc RNAs), methylation probes, miRNAs, and protein expression in the subgroups obtained by multi-omics clustering.

hallmark genesets to identify the genesets enriched in each ML cluster. As expected, the pathways associated with estrogen and androgen response were positively enriched in ML clusters C5 (Breast cluster) and C6 (Prostate cluster), respectively (S10 and S11 Tables).

Immunotherapy has become a crucial treatment option for many tumor types [50, 51]. The response to immunotherapy partially depends on the immune microenvironment of the tumor [52–54]. Hence, we analyzed the proportion of stromal (non-tumor) and leukocyte cells in each ML cluster (Fig 4). ML cluster C13 (Mixed) showed the highest infiltration of stromal and leukocyte cells (Fig 4(a) and 4(b)). To understand the proportion of leukocytes contributing to the stromal fraction, the leukocyte fraction was plotted against the stromal fraction (Fig 4(c)). In the case of the samples which are close to the diagonal or along the diagonal, leukocytes contribute to the majority of the stromal proportion [16]. Hence, C13 had an immune-rich microenvironment where the majority of the stromal fraction was contributed by the leukocytes.

To understand the difference in the immune microenvironment of each ML cluster, Cibersort analysis was performed using the LM22 geneset. We found that many clusters showed

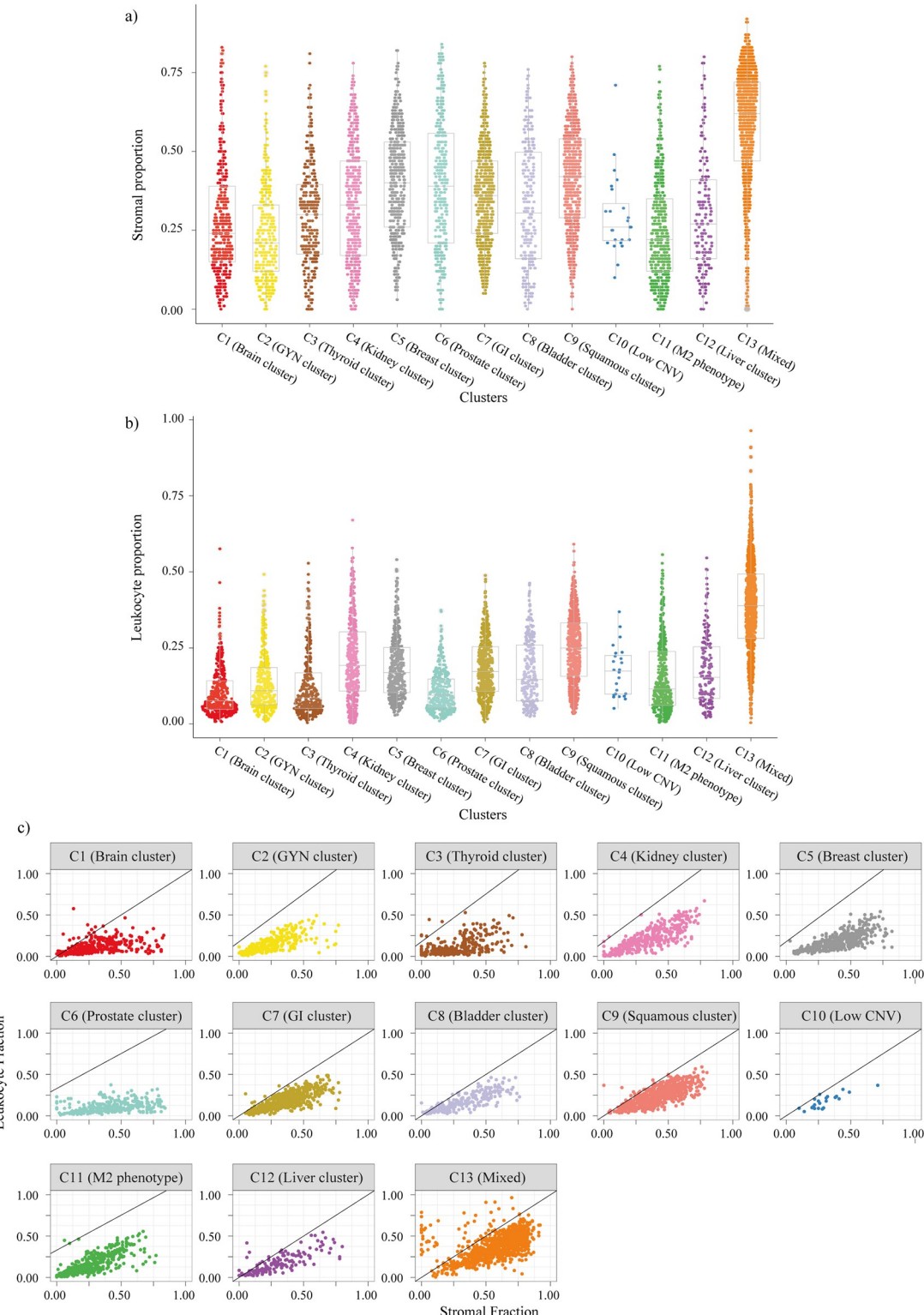

**Fig 4. Quantifying the tumor microenvironment of ML clusters. (a)** Stromal fraction for pan-cancer samples in each cluster. **(b)** Leukocyte fraction for pan-cancer samples in each cluster. **(c)** Leukocyte vs. Stromal fraction for pan-cancer samples in each cluster.

unique infiltration of various immune cells (S5 Fig and S12 Table). ML cluster C13 was observed to have a high infiltration of naive CD4 T cells which have a vital role to play in the immune response [55, 56]. ML cluster C11 had the highest infiltration of M2 macrophages and hence was named M2 phenotype.

As mutations have a significant role in tumor development and progression, non-silent mutation rates and mutation signatures were obtained for all the ML clusters (Fig 5). We found a significant variation in the mutation rates of each cluster. Interestingly, we found that ML cluster C8 (Bladder cluster) showed the highest mutation burden, and ML cluster C3 (Thyroid cluster) had the lowest median non-silent mutation rate (Fig 5(a)). The mutation signature obtained by non-negative matrix factorization (NMF) as described in the methods section, showed the different characteristics associated with each ML cluster

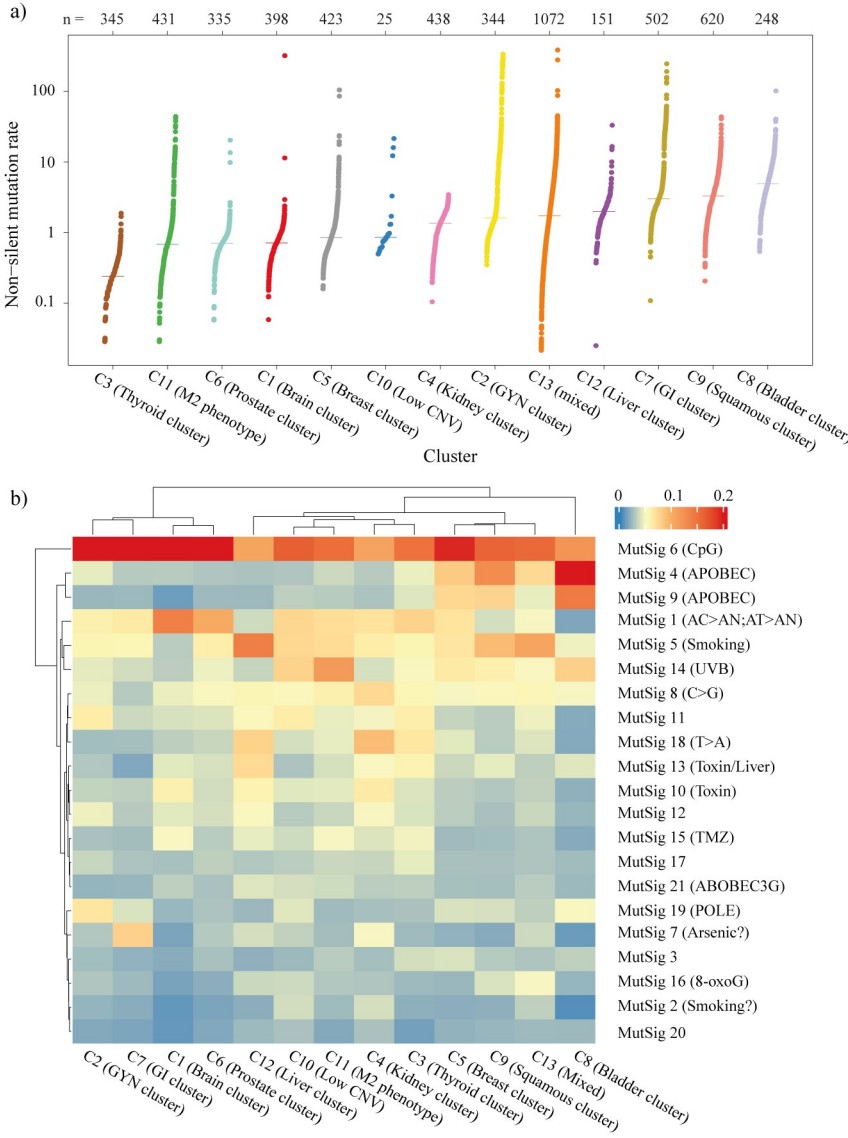

**Fig 5. Mutation analysis of ML clusters. (a)** Non-silent mutation in each cluster sorted by median. **(b)** Heatmap showing the average value of mutational signature in each cluster.

(Fig 5(b)). Signature 6 was the signature most commonly expressed across all the samples. It represents the CpG mutations indicating high enrichment of C>T mutations. ML cluster C2 (GYN cluster) had enrichment of MutSig 19 (POLE), indicating the activity of polymerase-E, which is generally associated with endometrial and colorectal cancers [57]. C2 was composed of UCEC, a subtype of endometrial carcinoma known to have POLE mutations. GI cluster (C7) was enriched for the mutation signature associated with the arsenic activity. APOBEC family of enzymes are known to cause mutation in various cancers [58]. We found enrichment of APOBEC mutation signature in the Bladder cluster (C8) and squamous cluster (C9) (Fig 5(b)). We also found enrichment of UVB mutation signature in ML cluster C11 which consists of SKCM and SARC cancers. ML cluster C13 (Mixed) with the majority of samples from LUAD (26%) had enrichment of mutation signature associated with smoking.

To understand the copy number variation in each ML cluster, the G-score obtained by GISTIC analysis was visualized using Maftools. The top five most frequently aberrated cytobands in each cluster were identified and plotted (S6 Fig). The analysis showed high CNV in ML cluster C2 (GYN cluster), C5 (Breast cluster), C7 (GI cluster), and C8 (Bladder cluster) and very minimal CNV in C3 (Thyroid cluster), C4 (Kidney cluster), C6 (Prostate cluster), and C10 (Low CNV).

Further, to interpret the pathway activity, the average value of the genes associated with 22 gene programs and 20 drug programs was plotted (S7 Fig) [16]. ML cluster C1 (Brain cluster) had the activity of gene programs associated with FOXO stemness, neural signaling, and tumor suppressing miRNA targets. ML clusters C2 (GYN cluster) and C7 (GI cluster) had higher expression of genes associated with cell-cell adhesion MClaudin cluster, C4 (Kidney cluster) of hypoxia glycolysis, C8 (Bladder cluster) of 1Q amplicon, and C13 (Mixed) of Immune T-cell B-cell. ML cluster C3 (Brain cluster) showed activation of the drug pathways associated with the ALK pathway and MYC amplified chr8q24. ML cluster C6 (Prostate cluster) had the activity of Nelson response to androgen up, and C12 (Liver cluster) of KEGG Retinol metabolism. Genes associated with Heller HDAC targets were expressed in ML clusters C8 (Bladder cluster) and C9 (Squamous cluster), and CTLA4 and PD1 signaling in ML clusters C4 (Kidney cluster) and C13 (Mixed).

## Subgroup identification by classifier combination

As all the studies do not have information from all the data types, classifiers were built using individual data types to identify the subgroup for a new sample that might have information from one data type only. Here, three classifiers, support vector machines (SVM), random forest (RF), and feed-forward neural network (FFNN), each based on different working principles, were trained. The prediction probabilities obtained from these classifiers were then combined using linear and non-linear models to obtain linear and non-linear decision-level fused models. All the decision-level fused models were obtained for data split with and without hold-out, as described in the methods section, to obtain better generalized models. The evaluation of different classifiers indicated the highest classification accuracy for the models built using $F_3$ (DNA methylation) data type, for both individual and decision-level fused models (Tables 2 and 3).

As each molecular level (data type) is known to carry and convey different and complementary information, the dimensions from each level were fused with the information from $F_3$ (as it had the highest classification accuracy) to obtain feature-level fused models. We did not observe a significant improvement in the accuracy of feature-level fused classification models as compared to individual classification models (Tables 2 and 3).

**Table 2. Summarizing the test accuracy of $L_0$ classifiers for different levels of evidence ($F_1$: mRNA expression, $F_2$: miRNA expression, $F_3$: DNA methylation, $F_4$: Protein expression, $F_{AE}$: Features from bottleneck layer of autoencoder, SVM: Support vector machine, RF: Random forest, FFNN: Feed-forward neural network).**

| Omic level | Dimension | w/o holdout | | | with holdout | | |
|---|---|---|---|---|---|---|---|
| | | SVM | RF | FFNN | SVM | RF | FFNN |
| $F_1$ | 3311 | 87.70 | 89.45 | 91.56 | 86.99 | 88.93 | 89.81 |
| $F_2$ | 104 | 86.29 | 76.10 | 84.18 | 84.53 | 76.45 | 80.49 |
| $F_3$ | 4872 | 93.85 | 95.43 | 96.66 | 92.79 | 94.90 | 92.97 |
| $F_4$ | 216 | 88.22 | 86.47 | 81.19 | 88.23 | 85.94 | 85.59 |
| $F_3 + F_1$ | 8183 | 80.49 | 94.20 | 95.70 | 74.69 | 94.73 | 96.48 |
| $F_3 + F_2$ | 4976 | 93.67 | 95.08 | 96.60 | 92.62 | 94.90 | 96.66 |
| $F_3 + F_4$ | 5088 | 93.32 | 94.90 | 95.60 | 92.27 | 94.03 | 97.01 |
| $F_{AE}$ | 100 | 98.59 | 97.54 | 97.89 | 97.19 | 97.01 | 95.78 |

**Table 3. Summarizing the test accuracy of $L_1$ classifiers for different levels of evidence ($F_1$: mRNA expression, $F_2$: miRNA expression, $F_3$: DNA methylation, $F_4$: Protein expression, $F_{AE}$: Features from bottleneck layer of autoencoder, LR: Logistic regression, FFNN: Feed-forward neural network).**

| Omic level | Dimension | w/o holdout | | | with holdout | | |
|---|---|---|---|---|---|---|---|
| | | Linear | LR | FFNN | Linear | LR | FFNN |
| $F_1$ | 3311 | 91.92 | 91.56 | 92.44 | 91.91 | 92.44 | 92.09 |
| $F_2$ | 104 | 76.10 | 82.07 | 75.22 | 85.41 | 86.29 | 83.66 |
| $F_3$ | 4872 | 97.72 | 97.36 | 97.36 | 96.48 | 96.66 | 96.66 |
| $F_4$ | 216 | 86.64 | 85.06 | 83.83 | 88.05 | 87.35 | 88.93 |
| $F_3 + F_1$ | 8183 | 96.31 | 94.38 | 95.60 | 96.13 | 95.96 | 95.61 |
| $F_3 + F_2$ | 4976 | 97.54 | 97.01 | 97.36 | 96.66 | 96.66 | 95.08 |
| $F_3 + F_4$ | 5088 | 96.66 | 96.66 | 96.31 | 97.72 | 97.19 | 96.48 |
| $F_{AE}$ | 100 | 98.95 | 98.77 | 98.77 | 97.89 | 97.36 | 95.43 |

To validate the classification models, the samples with methylation data that were not used to identify the multi-omics labels were chosen. The class labels for these samples were obtained using the previously trained classification models. Interestingly, the samples from Glioblastoma multiform (GBM) and ovarian cancer (OV), which were never used during training, were all assigned to C1 (Brain cluster) and C2 (GYN cluster), respectively (S8 Fig). Further, the gene and drug pathway analysis of these samples were obtained and validated (S9 Fig). It was observed that the variation in the pathway activity of validation samples was similar to that of the original samples. Hence, confirming the utility of the trained models to obtain the multi-omics labels even in the presence of a single datatype.

## Discussion

Despite the continuous improvements in the treatment strategies for cancer care, cancer still remains one of the leading causes of death worldwide. Though the existing treatment regimen has increased the overall survival time in some cancers, the five-year survival rate remains dismal for many cancer types. Current treatment strategies rely hugely on histopathological grades, which might not be the most appropriate factor in all cases as it is difficult to visually quantify the underlying molecular variations leading to the phenotypic change. Hence, with an aim to identify the subgroups beyond the histological subtypes, we integrated and analyzed information from various data types using machine learning (ML) models across the pan-cancer samples. In this work, information from mRNA, miRNA, DNA

methylation, and protein expression were considered for novel subgroup identification. For ML models to learn and recognize the patterns effectively, the condition, $n \sim p$ (where $n$ is the number of samples and $p$ is the number of dimensions) must be satisfied. Hence, we used variance-based filtering to ensure that the number of dimensions ($p$) is approximately equal to the number of samples ($n$).

To account for the non-linear interaction among the features, an autoencoder (AE) was trained, and the data was projected to a lower-dimensional space to identify the multi-omics-based molecular subgroups. Among the 13 molecular subgroups identified, six were pure, i.e., the samples were from the same organ-of-origin, and the rest were formed by the combination of samples from different tumor types (Fig 2(c)). Though the ML clusters C4 (Kidney cluster) and C7 (GI cluster) were formed by samples from different tumor types, the silhouette widths of these were comparable with pure Breast cluster (C5). This analysis revealed that molecularly similar samples from different tumor types could form a cluster that is as consistent as the cluster formed by the samples from a single tumor type.

Molecular analysis of the identified subgroups indicated similarities within the ML clusters in terms of various molecular aberrations despite the samples belonging to different tumor types. ML cluster C13 (Mixed) was the most diverse cluster with samples from 27 different tumor types. It also had the highest infiltration of stromal (non-tumoral) cells, leukocyte cells, and naive CD4 T cells (Fig 4 and S5 Fig). These findings suggest that patients from different tissue of origin are molecularly and immunologically similar, and can be considered for similar treatment regimens. ML cluster C8 (Bladder cluster) had the highest median non-silent mutation rate with enrichment for the APOBEC mutation signature, which is known to cause mutation in the bladder (Fig 5). ML cluster C3 (Thyriod cluster) had the lowest median non-silent mutation rate and enrichment of genes associated with thyroid hormone generation (Fig 5(a) and S6 Table).

Using the conventional treatment strategies, all the patients from the same tumor type would be treated using a similar approach. Following the proposed pipeline, samples from the same tumor type were split across multiple clusters, and various pathways enriched in each cluster can be used to target the samples (S6 and S10 Tables). LUAD (Lung adenocarcinoma) samples were divided into six different clusters (Fig 2(e) and S3 Fig). The GO pathways identified in each of these clusters indicated that they could be targeted using the inhibitors for microtubule bundle formation (C2—GYN cluster), epithelial cell differentiation (C7—GI cluster), epithelial cell differentiation (C8—Bladder cluster), epidermis development (C9—Squamous cluster), regulation of blood pressure (C10—Low CNV), and lymphocyte activation (C13—Mixed). CESC (Cervical squamous cell carcinoma and endocervical adenocarcinoma) and ESCA (Esophageal carcinoma) samples were split across four clusters C2 (GYN cluster), C7 (GI cluster), C9 (Squamous cluster), and C13 (Mixed) (Fig 2(e) and S3 Fig). The GO pathways that can be inhibited here include microtubule bundle formation (C2), epithelial cell differentiation (C7), epidermis development (C9), and lymphocyte activation (C13). ACC (Adrenocortical carcinoma), MESO (Mesothelioma), and PCPG (Pheochromocytoma and Paraganglioma) samples were split into two clusters C11 (M2 phenotype) and C13 (Mixed), which can be targeted using the inhibitors for the GO pathways phenol-containing compound biosynthetic process (C11), and lymphocyte activation (C13). C9 (Squamous cluster) had patients from 13 different tumor types (Fig 2(e) and S3 Fig). The samples here can be treated using the drugs targeting the positively enriched GSEA pathways, including, MTORC1 signaling, G2M checkpoint, E2F targets, MYC targets, or P53 pathway as summarised in S10 Table. C13 (Mixed) was comprised of samples from 27 different tumor types (Fig 2(e) and S3 Fig). The positively enriched GSEA pathways that could be targeted here include allograft rejection, IL6 JAK STAT3 signaling, inflammatory response,

interferon-gamma response, IL2 STAT5 signaling, TNFA signaling via NFKB, interferon-alpha response, KRAS signaling.

Different classification algorithms (SVM, RF, FFNN) were trained to identify the subgroup for a new incoming sample. The prediction probabilities from these classification models were combined to build decision-level fused models, as accounting for decisions from different classifiers helps in the reduction of variance in error, making the classification model more stable and robust to outliers. The models trained on methylation data ($F_3$) had the highest classification accuracy, indicating that this data type carried the highest information required for molecular subgroup identification (Tables 2 and 3). Classification models were also trained based on feature-level fusion techniques to account for the interaction among the molecular features from different data types. These models did not show a significant improvement in the classification accuracy when compared with the single datatype models. Though the combination of features adds additional information which will help in the classification, it will also lead to an increase in the dimension of the input data. The lack of improvement in the classification accuracy for the feature-level fused models might be because the classification models were trained with the same number of samples ($n$) but with an increased number of features ($p$). The model might not be able to capture and learn the pattern as $p \gg n$ in the case of the feature-level fused models.

Though the classification models were built separately for each data type, the class labels were obtained by multi-omics integrated data. We tested the proposed model on validation samples (samples with only $F_3$ data type) and proved that the proposed classification models could be used to identify the multi-omics molecular subgroup for a new sample even in the absence of multiple data types. Therefore, the subgroup identification technique proposed in this work might have clinical utility and provide additional information alongside the histological grades. Further, the molecular characteristics specific to each subgroup might help in the selection of an appropriate treatment strategy.

In this work, only a subset of features were chosen from each data type as part of the multi-omics data, which was then used for the identification of subgroups citing the limitation in terms of the number of samples. It is possible that incorporating additional features from the existing data types and also from other data types, like whole slide histopathological images, will add further information and provide more useful insights for a better understanding of cancer samples. This will be explored as part of our future work.

## Conclusion

This work attempted the identification of novel molecular subgroups in pan-cancer samples using multi-omics data. To handle the large dimensional multi-omics data and also to capture the non-linear interactions among the various datatypes, different machine learning-based techniques were applied. We identified 13 different subgroups independent of the organ-of-origin. We showed that the samples from different organs-of-origin can form clusters that are as stable as the clusters formed by the samples from similar organ-of-origin. Molecular characterization of the subgroups thus obtained highlighted the alterations specific to each subgroup, confirming the distinctness of each subgroup. An attempt was also made to build decision-level fused classification models to identify the subgroup for a new sample. Further, the classification models were validated using a single data type, and the molecular characteristics of samples were verified. Taken together, we applied ML-based methods on multi-omics data to identify the novel subgroups of cancer patients. Each subgroup was characterized to identify molecular changes at epigenetic and genetic levels. Also, classification models were built to classify an unseen sample in the 13 ML clusters.

## Supporting information

**S1 Fig. Steps followed in preprocessing each data type.**
(TIF)

**S2 Fig. CDF curves and KM plots for survival analysis.** (a) Figures showing the CDF curves for consensus clustering from $K = 2$ to 20. KM curves for (b) overall survival (OS) and (c) disease-free survival (DFS) in the ML clusters.
(TIF)

**S3 Fig. Figure showing the proportion of samples from each tumor type in each ML cluster.**
(TIF)

**S4 Fig. Steps followed in preprocessing each data type for molecular characterization.**
(TIF)

**S5 Fig. Heatmap showing the distribution of LM22 immune cells in each cluster.**
(TIF)

**S6 Fig. G-score of top 5 segments in each ML cluster indicating the copy number aberrations.** (a) to (m): ML cluster 1 to ML cluster 13.
(TIF)

**S7 Fig. Pathway analysis of ML clusters.** (a) Heatmap showing enrichment of different gene programs in all clusters. (b) Heatmap showing enrichment of different drug programs in all clusters.
(TIF)

**S8 Fig. Figure showing the proportion of samples from each tumor type in each ML cluster in the validation data set.**
(TIF)

**S9 Fig. Figure showing the activity of various programs in the validation data set.** (a) gene and (b) drug programs.
(TIF)

**S1 Table. Table describing the TCGA tumor types and the number of samples having information from different data types across different tumor types.**
(XLSX)

**S2 Table. Summarizing the data types, their mapping, number of samples, original dimension, and dimensions retained after preprocessing across pan-cancer samples.**
(XLSX)

**S3 Table. Training and validation losses for different architectures of AEs.**
(XLSX)

**S4 Table. Comparing the clustering scores between the clusters (K = 13) obtained using iClusterPlus and the proposed technique applied to the multi-omics data.**
(XLSX)

**S5 Table. Number of differentially expressed features in each datatype (PcG: Protein-coding genes, lnc RNA: Long non-coding RNA, miRNA: micro-RNA).**
(XLSX)

**S6 Table. Top five GO processes in each ML cluster identified by metascape analysis.**
(XLSX)

**S7 Table. Genes associated with the top five GO processes in each ML cluster identified by metascape analysis.**
(XLSX)

**S8 Table. Drug-gene interaction and druggable genes associated with each GO process in each ML cluster.**
(XLSX)

**S9 Table. Drug-gene interaction and druggable genes identified using the differentially expressed (DE) genes in each ML cluster.**
(XLSX)

**S10 Table. Positively enriched pathways in each subgroup obtained by GSEA analysis using hallmark geneset (FDR q≤0.05).**
(XLSX)

**S11 Table. Negatively enriched pathways in each subgroup obtained by GSEA analysis using hallmark geneset (FDR q≤0.05).**
(XLSX)

**S12 Table. Immune cells from LM22 gene signature enriched in each subgroup identified by Cibersort analysis.**
(XLSX)

**S1 File. The algorithm used to compute the values of $\alpha$, $\beta$, and $\gamma$ used to compute the prediction probabilities for the linear decision-level fused classification model.**
(PDF)

**S1 Graphical abstract.**
(TIF)

## Acknowledgments

The results shown here are in whole or part based upon data generated by the TCGA Research Network: https://www.cancer.gov/tcga.

## Author Contributions

**Conceptualization:** Seema Khadirnaikar, Sudhanshu Shukla, S. R. M. Prasanna.

**Formal analysis:** Seema Khadirnaikar.

**Investigation:** Sudhanshu Shukla.

**Methodology:** Seema Khadirnaikar, Sudhanshu Shukla.

**Resources:** Sudhanshu Shukla.

**Software:** Seema Khadirnaikar.

**Supervision:** Sudhanshu Shukla, S. R. M. Prasanna.

**Validation:** Seema Khadirnaikar.

**Visualization:** Seema Khadirnaikar.

**Writing – original draft:** Seema Khadirnaikar, Sudhanshu Shukla.

**Writing – review & editing:** Seema Khadirnaikar, Sudhanshu Shukla, S. R. M. Prasanna.

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
