## [Decision Letter · Decision Letter 0]

13 Feb 2023

PONE-D-22-28355Integration of Pan-cancer Multi-omics Data for Novel Mixed Subgroup Identification using Machine Learning MethodsPLOS ONE

Dear Dr. Shukla,

Thank you for submitting your manuscript to PLOS ONE. After careful consideration, we feel that it has merit but does not fully meet PLOS ONE’s publication criteria as it currently stands. Therefore, we invite you to submit a revised version of the manuscript that addresses the points raised during the review process.

Both the reviewers have asked some useful questions regarding the current manuscript. Kindly answer the same in the revised version.==============================

We look forward to receiving your revised manuscript.

Kind regards,

Sriparna Saha, PhD

Academic Editor

PLOS ONE

Journal Requirements:

“no”

Additional Editor Comments (if provided):

Both the reviewers have mentioned some comments on the paper. Authors should incorporate them and submit the revision.

Reviewers' comments:

Reviewer's Responses to Questions

**Comments to the Author**

1. Is the manuscript technically sound, and do the data support the conclusions?

Reviewer #1: Yes

Reviewer #2: No

2. Has the statistical analysis been performed appropriately and rigorously? 

Reviewer #1: I Don't Know

Reviewer #2: Yes

3. Have the authors made all data underlying the findings in their manuscript fully available?

Reviewer #1: Yes

Reviewer #2: No

4. Is the manuscript presented in an intelligible fashion and written in standard English?

Reviewer #1: Yes

Reviewer #2: Yes

5. Review Comments to the Author

Reviewer #1: In this paper, the authors are proposed “Integration of Pan-cancer Multi-omics Data for Novel Mixed Subgroup Identification using Machine Learning Methods”

The strengths of the paper are that it is well structured, the description of the related work is well done and that results are extensively compared to results of the similar research.

Minor revisions:

1. Authors should draw a graphical abstract of the proposed approach

2. Authors should justify the proposed approach

3. Authors write the comparison with other models developed for this cancer

4. Proofread the entire manuscript.

Reviewer #2: In this study the author have tried to integrate multi-omics studies to identify novel subgroup identification from different cancers. However the manuscript has multiple issues and there is overinterpretation of the data. Below are some of the points:

1. It is not clear how many samples were used for training and validation in the three methods SVM: support vector

machine, RF: random forest, FFNN: feed-forward neural network from each dataset and author need to mention this in the methods.

2. The authors say that the proposed framework can be used to identify 12 subgroups for new patients and help in personalized clinical decision making. Then this should performed on the another set of new samples as validation set.

3. Authors need to write in the discussion how these subgroups can be used to design appropriate treatment regimen for different cancers.

4. The authors have found GO process in each ML cluster and the genes identified in each GO process or pathway can be compared with The Drug Gene Interaction Database to identify clinically actionable gene targets identified in these clusters

5. The authors need to provide the list of genes identified in each GO process or pathway in these cluster as Supplementary table.

6. PLOS authors have the option to publish the peer review history of their article (what does this mean?). If published, this will include your full peer review and any attached files.

Reviewer #1: No

Reviewer #2: No

---

## [Author Response · Author response to Decision Letter 0]

20 Mar 2023

We would like to thank the editor and reviewers for their time and input. Please find the point-by-point response to all queries below.

Reviewer #1: 

1. Authors should draw a graphical abstract of the proposed approach

Response: We would like to thank the reviewer for the input. We have now included the graphical abstract of the proposed work (Reproduced here for the convenience of the reviewer). 

Rebuttal Figure 1: Graphical abstract of the proposed work is uploaded with the manuscript. 

2. Authors should justify the proposed approach

Response: We would like to thank the reviewer for the question and would like to apologize as this was not made clear in the manuscript.

Distinct phenotypes are the result of the underlying molecular variations in the genome. In cancer, the response to therapy relies hugely on the molecular alterations in the tumor. Hence, the stratification of patients based on the molecular characteristics of individuals has a significant role to play in tailoring treatment strategies to reduce adverse effects (Saria S et al. (2015), Vargas AJ et al. (2016)). Most of the existing studies proposed for subgroup identification are based on statistical techniques which might not be able to model the non-linear interactions occurring in the genome appropriately (Brigham & Women’s Hospital & Harvard Medical School et al. (2012), Mo Q et al. (2013), Hoadley KA et al. (2014), Wu D et al. (2015), Hoadley KA et al. (2018), Nguyen H et al. (2019), González-Reymúndez A et al. (2020)). However, the pipelines that are based on machine learning (ML) techniques have the ability to model non-linear interactions, but, the studies are limited to single tumor types (Chaudhary K et al. (2018), Baek B et al. (2020)). Tumors originating from different organs can have similar genetic and epigenetic variations. In order to identify the subgroups independent of the organ of origin, pan-cancer analysis which includes samples from all the tumor types, is carried out in this work. To obtain a holistic view of a heterogenous disease like cancer, there is an inherent need to model it using multiple levels of evidence, as they carry complementary information. To integrate and capture the non-linear interaction between the different levels of evidence, in this work, autoencoder (AE), an ML-based non-linear dimensionality reduction algorithm, is used. This reduced dimensional data is then clustered using consensus clustering to group the samples with similar characteristics together. 

To obtain further insights into the molecular features contributing to the similarity of the patients within the ML clusters, and the differences between the patients of different ML clusters, various statistical tests were carried out using different levels of evidence. To identify the pathways that can be targeted in each ML cluster, pathway analysis was carried out using the tools, GSEA, and metascape.

To identify the subgroup for a new sample, multiple classification models were built. The models were trained using each level of evidence separately, as most of the existing studies generate data by quantifying only one level of evidence. To propose a robust classification model, the prediction probabilities from these classifiers were combined to obtain decision-level fused classification models. 

Baek B, Lee H. Prediction of survival and recurrence in patients with pancreatic cancer by integrating multi-omics data. Scientific reports. 2020 Nov 3;10(1):18951.

Brigham & Women’s Hospital & Harvard Medical School Chin Lynda 9 11 Park Peter J. 12 Kucherlapati Raju 13, et al. "Comprehensive molecular portraits of human breast tumours." Nature 490.7418 (2012): 61-70.

Chaudhary K, Poirion OB, Lu L, Garmire LX. Deep learning–based multi-omics integration robustly predicts survival in liver Cancer using deep learning to predict liver cancer prognosis. Clinical Cancer Research. 2018 Mar 15;24(6):1248-59.

González-Reymúndez A, Vázquez AI. Multi-omic signatures identify pan-cancer classes of tumors beyond tissue of origin. Scientific reports. 2020 May 20;10(1):8341.

Hoadley KA, Yau C, Wolf DM, Cherniack AD, Tamborero D, Ng S, Leiserson MD, Niu B, McLellan MD, Uzunangelov V, Zhang J. Multiplatform analysis of 12 cancer types reveals molecular classification within and across tissues of origin. Cell. 2014 Aug 14;158(4):929-44.

Hoadley KA, Yau C, Hinoue T, Wolf DM, Lazar AJ, Drill E, Shen R, Taylor AM, Cherniack AD, Thorsson V, Akbani R. Cell-of-origin patterns dominate the molecular classification of 10,000 tumors from 33 types of cancer. Cell. 2018 Apr 5;173(2):291-304.

Mo Q, Wang S, Seshan VE, Olshen AB, Schultz N, Sander C, Powers RS, Ladanyi M, Shen R. Pattern discovery and cancer gene identification in integrated cancer genomic data. Proceedings of the National Academy of Sciences. 2013 Mar 12;110(11):4245-50.

Nguyen H, Shrestha S, Draghici S, Nguyen T. PINSPlus: a tool for tumor subtype discovery in integrated genomic data. Bioinformatics. 2019 Aug 15;35(16):2843-6.

Saria S, Goldenberg A. Subtyping: What it is and its role in precision medicine. IEEE Intelligent Systems. 2015 Jul 14;30(4):70-5.

Vargas AJ, Harris CC. Biomarker development in the precision medicine era: lung cancer as a case study. Nature Reviews Cancer. 2016 Aug;16(8):525-37.

Wu D, Wang D, Zhang MQ, Gu J. Fast dimension reduction and integrative clustering of multi-omics data using low-rank approximation: application to cancer molecular classification. BMC genomics. 2015 Dec;16(1):1-0.

3. Authors write the comparison with other models developed for this cancer

Response: We would like to thank the reviewer for the input.

Existing works on subgroup identification using pan-cancer data are based on statistical techniques. One of them is based on iClusterPlus (Hoadley et al. (2018)), and the other is on sparse singular value decomposition (sSVD) (González-Reymúndez A et al. (2020)). 

iClusterPlus is one of the widely used techniques to obtain subgroups using multi-omics data (Mo Q et al. (2013)). It is based on generalized linear regression. Here the authors model both discrete and continuous variables using appropriate random variables. A joint parametric model is then used to project this data to a latent space and identify the clusters. Hoadley et al. applied this technique to pan-cancer samples and identified 27 subgroups using methylation, mRNA expression, miRNA expression, and copy number variation (Hoadley et al. (2018)).

González-Reymúndez A et al. proposed a statistical technique based on sSVD to identify eight clusters using pan-cancer data (González-Reymúndez A et al. (2020)). The authors applied singular value decomposition (SVD) to identify the major axis of variation. This was penalized using the elastic net penalty to induce sparsity. The data projected along the major axis of variation was further reduced using t - Stochastic Neighbor Embedding (tSNE). This reduced dimension representation was clustered using Density-Based Spatial Clustering of Applications with Noise (DBSCAN) to identify the clusters. 

In contrast to the existing works, which are based on statistical techniques, the current work is based on ML algorithms for the identification of subgroups using multi-omics data. In the proposed work, data from mRNA expression, miRNA expression, DNA methylation, and protein expression are used. The multi-omics data from these levels of evidence is pre-processed, integrated, and projected to a lower dimension using AE, a non-linear dimensionality reduction technique. Consensus clustering is then carried out using this lower dimensional representation to identify the multi-omics subgroups. 

The clusters obtained using different pipelines are compared using two cluster evaluation techniques, silhouette coefficient, and Calinski-Harabasz index. The closer the value of the silhouette coefficient to one and the higher the Calinski-Harabasz index, the better the clustering. 

iClusterPlus was applied to the multi-omics data matrix obtained by concatenation of data from different data types. The parameters were tuned using tune.iClusterPlus as recommended by the authors (Hoadley et al. (2018), Mo Q et al. (2023)), and the pan-cancer patients were divided into 13 subgroups. Both the silhouette coefficient and the Calinski-Harabasz index indicated that the clusters obtained using the proposed algorithm were better separated than those obtained using the iClusterPlus (Rebuttal Table 1). 

To compare the proposed ML-based pipeline with the work of González-Reymúndez A et al., all the parameters were tuned as recommended by the authors (González-Reymúndez A et al. (2020)). Each data type was centered and scaled and then concatenated with other data types to obtain the multi-omics representation of the data. This was then projected to lower dimensional space representing 50 major axes of variation by sSVD. This data was further reduced to 2-dimensional space by tSNE and was followed by clustering using the DBSCAN algorithm. The resulting 15 clusters compared using the Silhouette coefficient and the Calinski-Harabasz index indicated better clusters using the proposed pipeline (Rebuttal Table 1). 

The pipelines based on statistical techniques will model the linear relationship appropriately but might not be able to capture the non-linear relationships. However, we know that the interaction between the different features in the genome is non-linear. We believe AE being a non-linear dimensionality reduction technique, was able to better model and capture this variation. Hence, resulted in better clusters as compared to other techniques (Rebuttal Table 1). 

Rebuttal Table 1: Comparing the clustering scores between the clusters obtained using iClusterPlus and sSVD followed by DBSCAN with the proposed technique applied to the multi-omics data.

Pipeline K Silhouette width Calinski-Harabasz index

iClusterPlus (Hoadley KA et al. (2018)) 13 0.06 414.03

sSVD + DBSCAN (González-Reymúndez A et al. (2020)) 15 -0.461 168.851

Proposed technique 13 0.24 565.31

González-Reymúndez A, Vázquez AI. Multi-omic signatures identify pan-cancer classes of tumors beyond tissue of origin. Scientific reports. 2020 May 20;10(1):8341.

Hoadley KA, Yau C, Hinoue T, Wolf DM, Lazar AJ, Drill E, Shen R, Taylor AM, Cherniack AD, Thorsson V, Akbani R. Cell-of-origin patterns dominate the molecular classification of 10,000 tumors from 33 types of cancer. Cell. 2018 Apr 5;173(2):291-304.

Mo Q, Wang S, Seshan VE, Olshen AB, Schultz N, Sander C, Powers RS, Ladanyi M, Shen R. Pattern discovery and cancer gene identification in integrated cancer genomic data. Proceedings of the National Academy of Sciences. 2013 Mar 12;110(11):4245-50.

Mo Q, Shen R (2023). iClusterPlus: Integrative clustering of multi-type genomic data. R package version 1.34.3.

4. Proofread the entire manuscript.

Response: We would like to apologize for the inconvenience caused. We have now thoroughly proofread the entire manuscript and made the appropriate corrections.

Reviewer #2: 

1. It is not clear how many samples were used for training and validation in the three methods SVM: support vector machine, RF: random forest, FFNN: feed-forward neural network from each dataset and author need to mention this in the methods.

Response: We would like to apologize for the inconvenience caused. 

The total number of samples used to carry out the analysis was N = 5703. The train-test split of 90%-10% was used to obtain the train (N = 5132) and test (N = 571) data. The training data was further split (90%-10%) to obtain the train (N = 4619) and validation (N = 513) set. 

We have trained three ML models, SVM, RF, and FFNN (L0). To obtain a robust classifier, the prediction probabilities from these classifiers were combined, and a decision-level fused classifier (L1) was obtained. We have trained two types of decision-level fused classifiers, one without a hold-out dataset and the other one with a hold-out dataset. In the decision-level fused classifier without hold-out, both L0 and L1 levels of classifiers were trained using all the training samples (N = 4619). For the decision-level fused classifier with hold-out, a hold-out dataset is created by splitting the training dataset in the ratio of 60%-40%. The L0 model is trained using 60% (N = 2771) and L1 using 40% (N = 1848) of the data from the training set. 

We have now added these details to the manuscript.

 2. The authors say that the proposed framework can be used to identify 12 subgroups for new patients and help in personalized clinical decision making. Then this should performed on the another set of new samples as validation set.

Response: We would like to thank the reviewer for the input. We understand that the proposed pipeline must be validated in an independent dataset. We are actively looking for a dataset with information from all levels of evidence and clinical data to validate the proposed pipeline. However, we would like to mention that the model was validated using the TCGA samples, which had only methylation values and were not used to obtain the cluster labels or to build the classification models (S8 and S9 Figure). We do understand this does not answer the question adequately, and validation of the proposed model in an independent dataset will be considered as part of future work.

3. Authors need to write in the discussion how these subgroups can be used to design appropriate treatment regimen for different cancers.

Response: We would like to thank the reviewer for the input. 

Using the conventional treatment strategies, all the patients from the same tumor type would be treated using a similar approach. Following the proposed pipeline, samples from the same tumor type were split across multiple clusters, and various pathways enriched in each cluster can be used to target the samples. The pathways enriched in different clusters were identified using metascape analysis (GO pathways, S6 Table) and GSEA analysis (Hallmark geneset, S10 Table).

LUAD (Lung adenocarcinoma) samples were divided into six different clusters (Fig. 2 (e) and S3 Fig). The GO pathways identified in each of these clusters indicated that they could be targeted using the inhibitors for microtubule bundle formation (C2 - GYN cluster), epithelial cell differentiation (C7 - GI cluster), epithelial cell differentiation (C8 - Bladder cluster), epidermis development (C9 - Squamous cluster), regulation of blood pressure (C10 - Low CNV), and lymphocyte activation (C13 - Mixed).

CESC (Cervical squamous cell carcinoma and endocervical adenocarcinoma) and ESCA (Esophageal carcinoma) samples were split across four clusters C2 (GYN cluster), C7 (GI cluster), C9 (Squamous cluster), and C13 (Mixed) (Fig. 2 (e) and S3 Fig). The GO pathways that can be inhibited here include microtubule bundle formation (C2), epithelial cell differentiation (C7), epidermis development (C9), and lymphocyte activation (C13).

ACC (Adrenocortical carcinoma), MESO (Mesothelioma), and PCPG (Pheochromocytoma and Paraganglioma) samples were split into two clusters C11 (M2 phenotype) and C13 (Mixed), which can be targeted using the inhibitors for the GO pathways phenol-containing compound biosynthetic process (C11), and lymphocyte activation (C13).

C9 (Squamous cluster) had patients from 13 different tumor types (Fig. 2 (e) and S3 Fig). The samples here can be treated using the drugs targeting the positively enriched GSEA pathways, including, MTORC1 signaling, G2M checkpoint, E2F targets, MYC targets, or P53 pathway as summarised in S10 Table.

C13 (Mixed) was comprised of samples from 27 different tumor types (Fig. 2 (e) and S3 Fig). The positively enriched GSEA pathways that could be targeted here include allograft rejection, IL6 JAK STAT3 signaling, inflammatory response, interferon-gamma response, IL2 STAT5 signaling, TNFA signaling via NFKB, interferon-alpha response, KRAS signaling. 

These details are now updated in the discussion section of the manuscript.

4. The authors have found GO process in each ML cluster and the genes identified in each GO process or pathway can be compared with The Drug Gene Interaction Database to identify clinically actionable gene targets identified in these clusters

Response: We would like to thank the reviewer for the input. We had summarized the top five GO pathways associated with each cluster in S6 Table. We have now used the gene list obtained from each GO pathway to identify the drug-gene interaction and druggable genes in each GO process (Freshour et al. (2021)). These results are now added as part of the supplementary table (S8 Table). 

We have also carried out the drug-gene interaction analysis using the DE genes in each cluster. These results are now included in S9 Table.

Freshour, Sharon L., et al. "Integration of the Drug–Gene Interaction Database (DGIdb 4.0) with open crowdsource efforts." Nucleic acids research 49.D1 (2021): D1144-D1151.

5. The authors need to provide the list of genes identified in each GO process or pathway in these cluster as Supplementary table.

Response: We would like to apologize for the inconvenience caused. The differentially expressed (DE) protein-coding genes (PcGs) in each cluster were used to carry out the metascape analysis. The resulting top five GO pathways were summarized in S6 Table. We have now included the gene list associated with each GO pathway as the supplementary table (S7 Table).

---

## [Decision Letter · Decision Letter 1]

1 Jun 2023

Integration of Pan-cancer Multi-omics Data for Novel Mixed Subgroup Identification using Machine Learning Methods

PONE-D-22-28355R1

Dear Dr. Shukla,

We’re pleased to inform you that your manuscript has been judged scientifically suitable for publication and will be formally accepted for publication once it meets all outstanding technical requirements.

Kind regards,

Amy McCart Reed

Academic Editor

PLOS ONE

Additional Editor Comments (optional):

Thank you for addressing all the comments made by the reviewers.

Reviewers' comments:

Reviewer's Responses to Questions

**Comments to the Author**

1. If the authors have adequately addressed your comments raised in a previous round of review and you feel that this manuscript is now acceptable for publication, you may indicate that here to bypass the “Comments to the Author” section, enter your conflict of interest statement in the “Confidential to Editor” section, and submit your "Accept" recommendation.

Reviewer #2: All comments have been addressed

Reviewer #3: All comments have been addressed

2. Is the manuscript technically sound, and do the data support the conclusions?

Reviewer #2: Yes

Reviewer #3: Yes

3. Has the statistical analysis been performed appropriately and rigorously? 

Reviewer #2: Yes

Reviewer #3: I Don't Know

4. Have the authors made all data underlying the findings in their manuscript fully available?

Reviewer #2: Yes

Reviewer #3: Yes

5. Is the manuscript presented in an intelligible fashion and written in standard English?

Reviewer #2: Yes

Reviewer #3: Yes

6. Review Comments to the Author

Reviewer #2: (No Response)

Reviewer #3: This is a useful and interesting study to combine pan-cancer cohorts across multiple omics dimensions and then co-clustering after dimensionality reduction to identify pan-cancer clusters. The tumours within each of the clusters, despite coming from different cancer types, have common molecular characteristics. This will be useful to design basket trials for drugs based on molecular composition rather that just cancer type.

I do not have any further comments. The authors have addressed the comments from the earlier reviewers satisfactorily.

7. PLOS authors have the option to publish the peer review history of their article (what does this mean?). If published, this will include your full peer review and any attached files.

Reviewer #2: No

Reviewer #3: No

---

## [Editor Report · Acceptance letter]

12 Jun 2023

PONE-D-22-28355R1 

Integration of Pan-cancer Multi-omics Data for Novel Mixed Subgroup Identification using Machine Learning Methods 

Dear Dr. Shukla:

I'm pleased to inform you that your manuscript has been deemed suitable for publication in PLOS ONE. Congratulations! Your manuscript is now with our production department. 

Kind regards, 

on behalf of

Dr. Amy McCart Reed 

Academic Editor

PLOS ONE